

**Global Stable Isotope Dataset for Surface Water**
Rui Li[1,2,3], Guofeng Zhu[1,2,3*], Longhu Chen[1,2,3], Xiaoyu Qi[1], Siyu Lu[1,2,3], Gaojia Meng[1,2,3], Yuhao
Wang[1,2,3], Wenmin Li[1,2,3], Zhijie Zheng[1,2,3] , Jiangwei Yang[1,2,3] , Yani Gun[1,2,3]
**Affiliations:**
*[1] College of Geography and Environmental Science, Northwest Normal University, Lanzhou*
*730070, Gansu, China*
*[2] Shiyang River Ecological Environment Observation Station, Northwest Normal University,*
*Lanzhou 730070, Gansu, China*
*[3] Key Laboratory of Resource Environment and Sustainable Development of Oasis, Gansu*
*Province, Lanzhou 730070, Gansu, China*
*Correspondence to: zhugf@nwnu.edu.cn*
**Abstract:** Hydrogen and oxygen-stable isotopes are widely used as tracers of the water cycle, and
surface water is an integral part of the water cycle. Compared with other water bodies, surface
water is more susceptible to different natural and anthropogenic factors, and an accurate
understanding of surface water changes is of great significance in ensuring regional water security,
maintaining ecological balance, and promoting sustainable economic and social development. Due
to the influence of observation conditions and instrumental analysis, global surface water isotopes'
spatial and temporal distribution could be more balanced worldwide. For this reason, we have
compiled and analyzed the stable hydrogen and oxygen isotope data in surface water from 22432
sampling stations worldwide from 1956 to 2023, with 102862 data records. The results found: (1)
global surface water stable isotopes are gradually depleted from the equator to the poles and from
the coast to the interior. However, there are significant differences in the spatial and temporal
distributions of surface water isotopes in different regions. (2) The variation of stable isotopes in
surface water is controlled by geographic location, topographic conditions, and meteorological
factors (especially temperature), and its heterogeneity is considerable. The global stable isotope
dataset of surface water provides vital information for an in-depth understanding of the water
cycle and climate change. It can provide essential data references for global water resource
management and research. The Global Surface Water Stable Isotope Dataset is available at



https://doi.org/10.17632/fs7rwp7fpr.1 (Zhu, 2024).
**Keywords:** Stable isotopes; Surface Water; Global Dataset

## 1.Introduction

Water resources are an essential material basis for human survival and indispensable for
maintaining sustainable local socio-economic development, preserving ecological health, and
maintaining ecosystem stability (Immerzeel et al., 2020; Mehta et al., 2024). Due to human
activity and climate change, global hydrological systems have changed in recent decades,
increasing ecological vulnerability and sensitivity to climate change (Chahine, 1992; Liu et al.,
2021; Satoh et al., 2022). Hydrogen and oxygen isotopes, as a kind of stable isotopes widely
present in the water column (Reckerth et al., 2017; Sprenger et al., 2016), are an important method
for conducting water cycle studies and have an essential indicative role in the study of the water
cycle (Aggarwal et al., 2007; Joussaume et al., 1984; Vystavna et al., 2021). However, due to
restrictions imposed by their conditions in various regions of the world, there are a number of
difficulties and constraints in the gathering, integrating, and analysing of current stable isotope
data for surface water (Chen et al., 2020; Penna et al., 2018).
Since 1960, the Global Network of Isotopes in Precipitation (GNIP) was created by the
International Atomic Energy Agency (IAEA) and the World Meteorological Organization (WMO),
with the aim of constructing a worldwide monitoring network focusing on the in-depth study of
hydrogen and oxygen isotopes in precipitation (Aggarwal et al., 2012). Global surface water
monitoring networks have developed more slowly than stable isotope monitoring of precipitation.
In 2002, the IAEA started building the Global Network of Isotopes in Rivers (GNIR), which aims
to study the interactions between surface water and groundwater using stable isotopes in runoff
and to identify the effects of climate change on river runoff and the effects of human activity on
riverine variability (Halder et al., 2015). Many academics worldwide have studied the stable
isotope composition of surface water, which is influenced by a range of hydrological processes
like precipitation, evaporation, melting, and surface runoff. This composition can provide
important insights into the functioning of the water cycle, the management of water resources, and
the effects of climate chang (Bowen et al., 2019; Darling, 2004; Schulte et al., 2011). The source,
flow, accumulation, and change rule of surface water can be thoroughly understood by analysing





and interpreting stable isotope data, which can offer a scientific foundation for water resource
management, water resource assessment, and ecological and environmental protection. In addition,
surface water, as a "link" between groundwater and precipitation (Cooley et al., 2021), offers fresh
scientific perspectives on a variety of hydrogeological phenomena, including the hydrogeologic
evolution of the basin (Bershaw et al., 2016), groundwater-surfac water interactions(Autio et al.,
2023), groundwater recharge (Jameel et al., 2023), and precipitation processes (Bershaw et al.,

65   2016).

In light of global climate change and water scarcity, creating a global stable isotope dataset
for surface water is important.   The creation of a global stable isotope dataset for surface water
will facilitate the utilisation and integration of surface water isotope data resources across various
regions, enhance data accessibility and usability, and offer researchers more dependable and
abundant data support for conducting global hydrological and environmental studies. In the
meanwhile, studies on water resource assessment, climate change adaptation, and agricultural
irrigation optimisation can be carried out using the global surface water stable isotope dataset.
These studies can offer a scientific foundation for resolving important problems in the
management of water resources globally. In this work, we present the first global surface water
stable isotope dataset, comprising measured, website, and references data. Our goals are as
follows: The goals are as follows: (1) to compile and gather surface water stable isotope data
globally; (2) to construct a global surface water stable isotope dataset, and to promote the
application of global surface water stable isotope dataset in the hydrological, meteorological,
ecological, and other fields.
**2.  Data and methods**
**2.1 Composition of the dataset**
The Dataset consists of three main elements: website data (GNIR data, http://nucleus.i
aea.org/wiser/explore, water isotopes website, http://wateriso.utah.edu/waterisotopes), measure
d data and references data. The dataset encompasses 22432 surface water sampling sites a
cross seven continents (Fig.1). Since 2015, an ecohydrological observation system has been
implemented in the Shiyang River Basin in the arid zone of Northwest China to systema
tically gather surface water stable isotope data, serving as the primary source of measured



data.

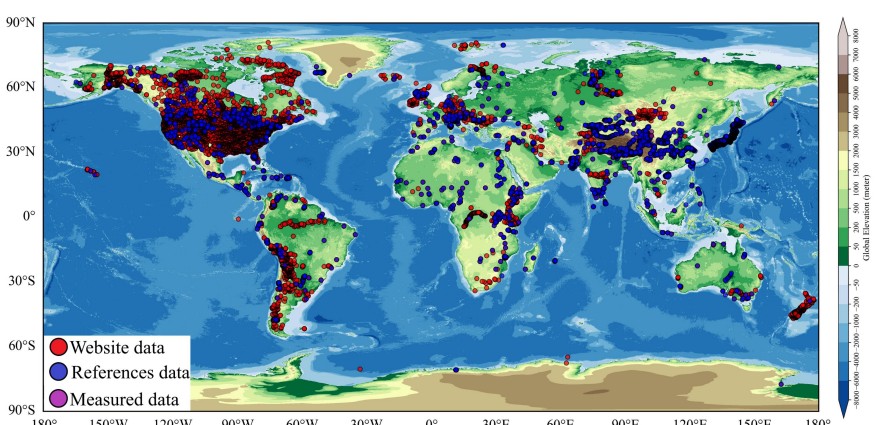

**Figure 1** Distribution of sample sites in the global stable isotope dataset for surface water.
Measured data: Surface water sampling sites are chosen whenever feasible at places where
the water is moving quickly because stagnant water is frequently impacted by pollution and
evaporation. After the sampling bottle was rinsed three times prior to sampling using water from
the sampling site, the bottle was placed below the surface of the water with the mouth facing up
and filled to a position approximately three-quarters of the bottle's volume. Following the
completion of the water sample collection process, the bottles are promptly sealed tightly, their
mouths are taped with waterproof tape, and labels bearing the name of the sampling location, the
sampling date, and additional information are affixed to the bottles. Every collected water sample
was kept in a refrigerator to be frozen in order to avoid data errors caused by evaporation.
References data: We added more information to the database by searching for the terms
"isotope," "surface water," and "river" in published papers on Web-of-Science. We chose scholarly
articles containing isotope data in textual, tabular, and graphical formats as the primary source of
data to enhance the precision of our data. The aforementioned papers explicitly identified the
water body type as "surface water." Alongside isotope data, we gathered spatial and temporal
informations, including the latitude and longitude of the sampling sites and the exact time of
sampling.
Moreover, the meteorological data utilized in this study were sourced from the NCEP-
NCAR reanalysis dataset (https://psl.noaa.gov/data/gridded/data.ncep.reanalysis.html) and the
CRUTS v. 4.07 dataset (https://crudata.uea.ac.uk/cru/data). The data utilized for the global



climate division are derived from Köppen's global climate classification (Peel et al., 2007)
(Fig. S1).

## 2.2 Data processing

Prior to the experiment commencing, removing the samples to be analysed from the
refrigerator and transferring them to standard 1.5 mL glass sample bottles once they had melted in
the room. A filter with a pore size of 0.45 μm and a diameter of 13 mm was then applied to
eliminate any contaminants, such as silt and dust, that may have been carried in with the samples
during the transfer. All water samples were analyzed for stable isotope values using a liquid water
isotope analyzer (DLT-100, Los Gatos Research, USA). During the determination process, each
water sample was measured six consecutive times. To prevent residual contamination from
affecting the results, the first two measurements were discarded, and the stable isotope value was
calculated as the average of the last four measurements. The test results obtained are expressed as
thousandths deviation from the Vienna Standard Mean Ocean Water (V-SMOW):

$$\delta_{\text{sample}}(‰) = [(\frac{R_s}{R_v}) - 1] \times 1000$$

Here, $R_s$ represents the ratio of $^{18}O/^{16}O$ or $^2H/^1H$ in the collected sample, and $R_v$ is the ratio of
$^{18}O/^{16}O$ or $^2H/^1H$ in the Vienna standard sample. The analytical accuracy for δD and δ$^{18}$O is
±0.6‰ and ±0.2‰, respectively.
To ensure data accuracy, we used LIMA to test the raw data generated by the analyzer. Only
data that passed the software test were included in the dataset. If the data did not pass, the analysis
was repeated until it did. Additionally, all isotope data were thoroughly examined to ensure each
entry included clear "longitude," "latitude," "sampling time," and "isotope" data. Outliers and
duplicates were removed (Fig. 2).





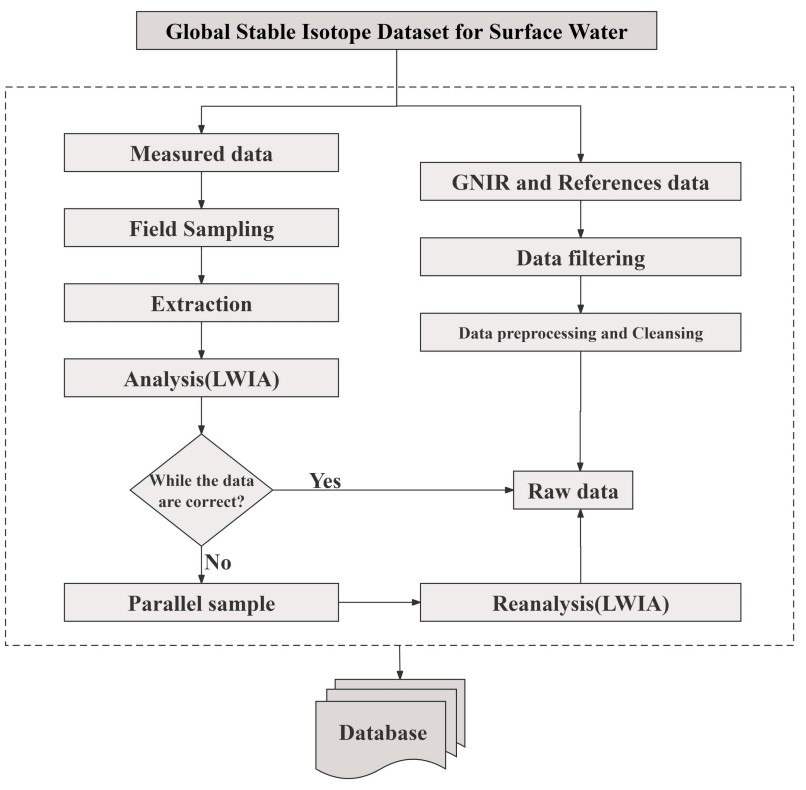


**Figure 2** Flow of data processing and construction of global surface water stable isotope dataset.
**2.3 Methods**
Based on previous studies, a one-way ANOVA was used to determine the significance ($p <$
0.05 at a 95% confidence level) of the slopes and intercepts of the linear regression fits for surface
water stable isotopes $\delta D$ and $\delta^{18}O$ across different climatic regions (Vystavna et al., 2021).
Furthermore, the Random Forest (RF) model can assess the importance of variables. In this study,
we employed the Random Forest model to evaluate the impact of various meteorological factors
on the stable isotopes of surface water globally. The Random Forest algorithm integrates multiple
decision trees to generate a cumulative effect. It predicts regression outcomes based on the
average results of these randomized decision trees, employing bootstrapping to minimize the risk
of overfitting (Breiman, 2001; Hu et al., 2017). Both root mean square error (RMSE) and mean
absolute error (MAE) were utilized to estimate the model's error (Kartal, 2024).



## 3. Results and discussions

### 3.1 Volume, geographic distribution and temporal coverage of datasets

As shown in Fig. 3, a total of 102862 measurements of stable isotopes of hydrogen and oxygen in surface water were collected for this dataset. This includes 79525 website data, 1101 measured data, and 22236 references data. Most of the GNIR data are concentrated in a few regions, such as the United States and Eastern Europe, resulting in a sparse global distribution with regional concentrations. GNIR data are primarily concentrated in a few regions, such as the United States and Eastern Europe, and are sparsely distributed globally. To expand our dataset, we incorporated data from published literature. This expanded dataset now covers nearly the entire world with a relatively even distribution, including regions traditionally difficult to access data from, such as Greenland, Antarctica, western Australia, and high-altitude mountainous areas (Fig. 1). The dataset spans from 1956 to 2023, with the majority of data collected from 1990 onwards. This timeframe indicates that the dataset effectively captures the global distribution characteristics of stable isotopes in surface water over the past few decades.

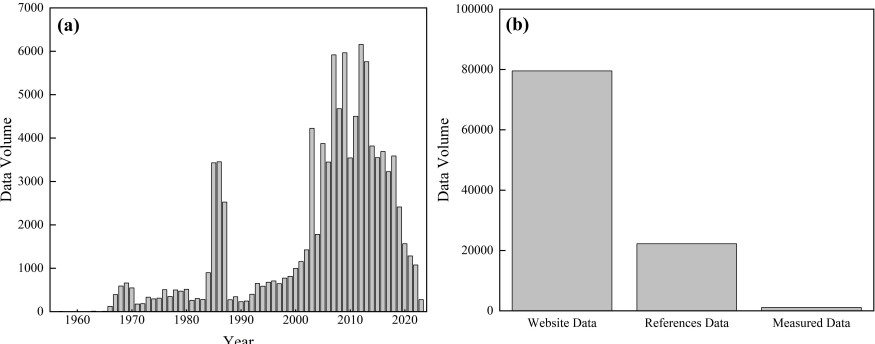

**Figure 3** Distribution of global surface water stable isotope dataset. (a) time series distribution; (b) category distribution.

### 3.2 Spatial and temporal variations of stable isotopes in global surface waters

The variation of $\delta D$ ranged from -340.85‰ to 74.01‰, and $\delta^{18}O$ ranged from -42.30‰ to 20.41‰ over the whole dataset. On a seasonal scale, global surface water stable isotopes typically exhibit pronounced variations, characterized by higher values in summer and lower values in winter. To better observe these variations across different regions, we classified the globe into five climatic zones—tropical, temperate, arid, continental, and polar, based on the "Köppen climate

zones" classification. Across the six climatic zones, stable isotopes of surface water exhibit
seasonal variations with higher values in summer and lower values in winter, except in polar
climatic zones.    The most pronounced variations occur in arid zones, underscoring the influence
of climatic factors on stable isotopes of surface water.

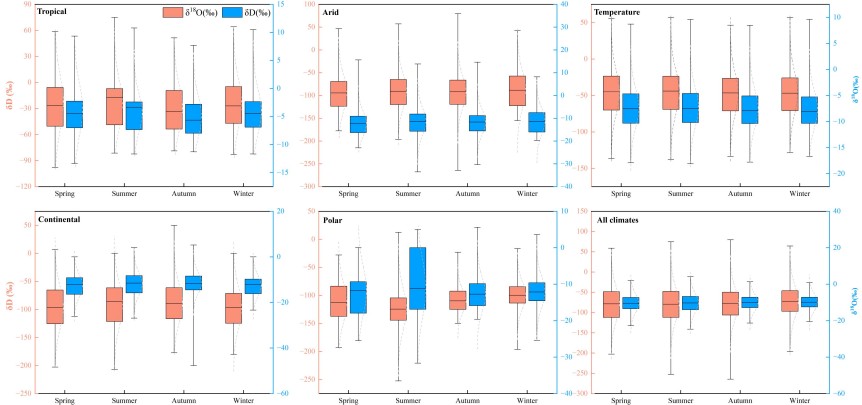

**Figure 4** Seasonal variation of δD and δ¹⁸O in surface water in different climatic zones.

Meanwhile, to better describe the spatial distribution of stable isotopes in global surface

water, we conducted interpolation to map their spatial distribution globally (Fig. 5). Generally, δD
and δ¹⁸O exhibit a consistent trend of gradually decreasing values from equatorial regions to high
latitudes and from coastal regions to inland areas of continents such as Eurasia and North America.
This trend is especially pronounced in high-latitude and high-altitude regions, where the values are
significantly lower. However, some areas do not exhibit a clear pattern in the distribution of δD
and δ¹⁸O values. This irregularity primarily results from the complex factors influencing runoff
generation and water flow concentration processes in various regions. Additionally, the presence
of open water bodies, such as lakes and reservoirs, exacerbates this irregular distribution
phenomenon.

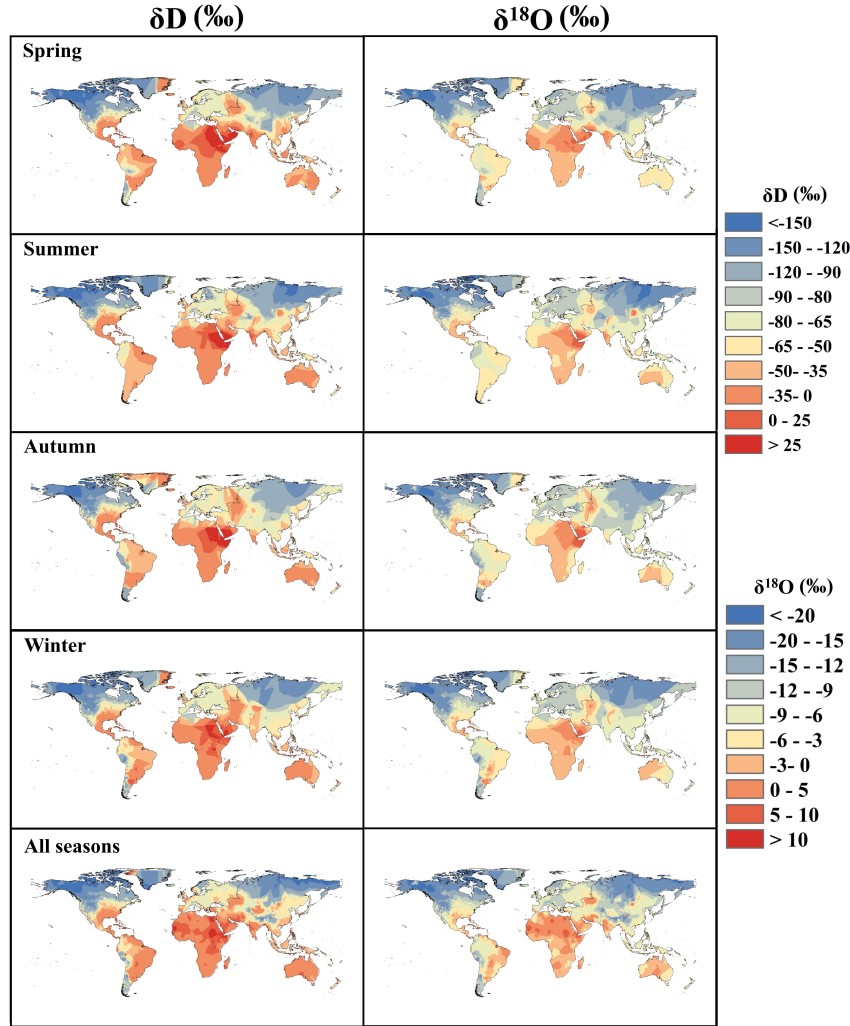

**Figure 5** Spatial distribution of global surface water δD and δ<sup>18</sup>O in different seasons.

To better understand the relationship between surface water and precipitation, we compared the spatial interpolation results of surface water isotopes with those of global precipitation isotopes. We found that the isotope distribution in surface water is largely consistent with the isotope distribution in precipitation across most areas (Fig. S2). This consistency primarily arises because surface water is predominantly recharged by precipitation. Moreover, the spatial variation in the isotopic composition of surface water serves as a valuable indicator of its recharge relationship with groundwater and precipitation (Kendall and Coplen, 2001). This is particularly evident in the tropics and at high altitudes, where precipitation serves as the primary source of



surface water recharge. In these regions, the spatial distributions of surface water isotopes and
precipitation isotopes exhibit a high degree of similarity (Fig. S2).

### 3.3 Controlling factors for stable isotopes in surface water

For precipitation stable isotopes, there is a significant "latitude effect" and "continent effect
(Dansgaard, 1964)," this pattern of variation is also observed in the stable isotopes of surface
water, characterized by a gradual decrease in stable isotope values from low to high latitudes and
from coastal to arid inland areas. However, in low-latitude regions near the equator, where surface
water is primarily recharged by precipitation and climatic factors do not vary significantly along
latitude, there is no significant spatial variation in the stable isotopes of surface water.
Additionally, numerous studies have demonstrated that the stable isotope composition of
surface water is predominantly influenced by climatic factors (Araguás-Araguás et al., 1998;
Dansgaard, 1964; Wang et al., 2017). To assess the importance of various meteorological variables
on the stable isotopes of surface water globally, we employed a RF model. The RF regression
analysis fitted to the stable isotopes of surface water indicated a strong model fit for both the
training and test sets. This suggests that variables such as temperature, precipitation, potential
evapotranspiration, vapor pressure, wind speed, and relative humidity possess significant
explanatory power for the stable isotopes of surface water (Fig. 6). The validation results of the
RF model demonstrate excellent prediction performance for both $\delta^{18}$O and $\delta$D, with $\delta^{18}$O showing
better prediction accuracy than $\delta$D, as indicated by smaller RMSE and MAE values (Table S1).
Among the six meteorological factors considered, temperature exerts the strongest influence on
surface water stable isotopes. Potential evapotranspiration also exhibits a strong controlling effect,
suggesting that temperature and evapotranspiration are the primary factors governing changes in
global surface water stable isotopes. Additionally, relative humidity and wind speed demonstrate
high explanatory power for variations in surface water stable isotopes. Previous studies have
indicated that wind speed and relative humidity significantly influence evaporation from water
bodies (Gallart et al., 2024; Skrzypek et al., 2015), which can subsequently impact surface water
stable isotopes. While vapor pressure and precipitation offer weaker explanations for variations in
surface water stable isotopes, these factors can largely be attributed to the residence time of
surface water and the local hydrological cycle. The residence time of surface water and the
characteristics of the local hydrological cycle vary significantly across different regions. Large
open water bodies typically have longer residence times and slower hydrological cycles, resulting
in a more enriched isotopic composition of surface water (Feng et al., 2016). In contrast, water
bodies with faster hydrological cycles, such as rivers, may exhibit different isotopic compositions
(Ala-aho et al., 2018). However, interpreting these patterns on a large scale requires further
investigation and validation.

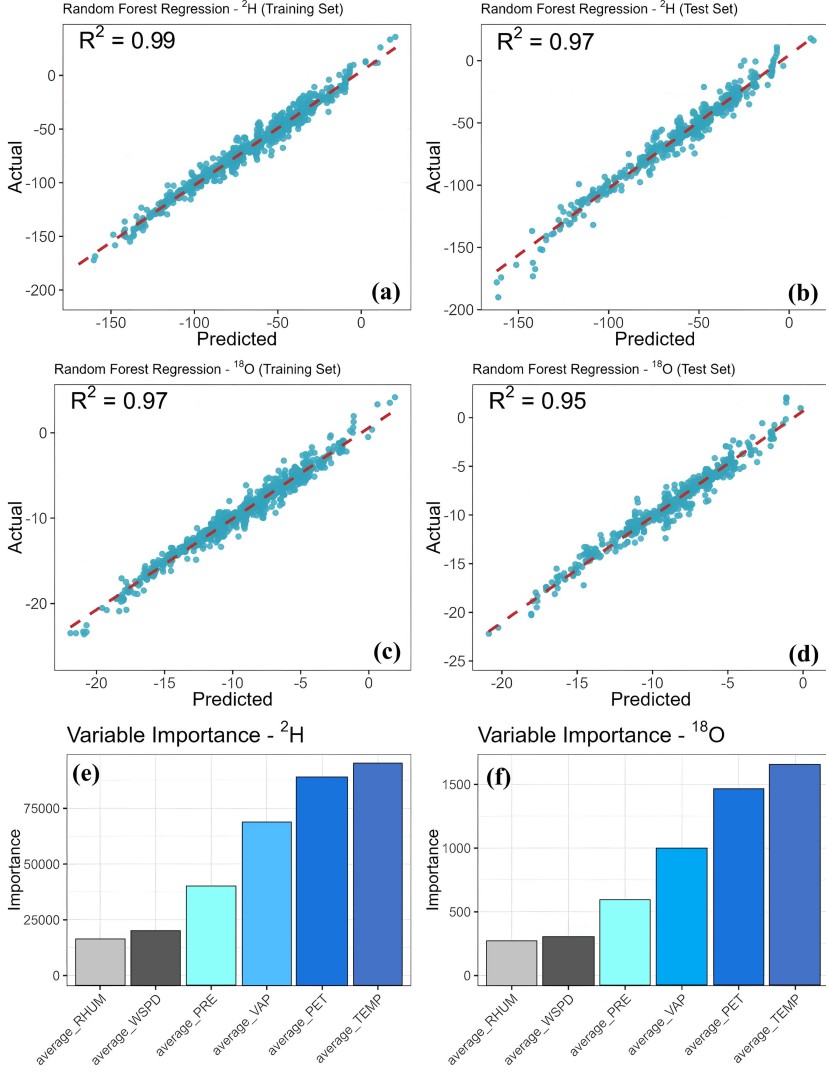


**Figure 6** The relationship between δD and δ¹⁸O and meteorological factors was analyzed using RF
model. (a) δD regression results for the training set. (b) δD regression results of the test set. (c)
δ¹⁸O regression results of the training set. (d) δ¹⁸O regression results of the test set. (e) Effect of



meteorological factors on δD. (f) Effect of meteorological factors on δ¹⁸O.
Simultaneously, for lakes, reservoirs, and other large open water bodies, the controls on
surface water stable isotopes can be more complex. Studies have demonstrated that global stable
isotope variations in lakes result from the combined effects of solar radiation, evapotranspiration,
catchment area size, and other factors (Vystavna et al., 2021). These controls vary across different
regions, contributing to diverse stable isotopic compositions in surface waters worldwide. For
instance, in arid zones, solar radiation primarily controls stable isotopic variations in lakes,
whereas in temperate climatic zones, evaporation and transpiration play a dominant role.
Consequently, the controlling factors for surface water stable isotopes vary significantly across
different regions. However, overarching patterns suggest that geographic and meteorological
factors collectively govern the stable isotopic changes in surface water within a region.
**3.4 Contribution of global surface water stable isotope datasets to the**
**understanding of the global water cycle, climate change and ecosystem processes**
In recent decades, stable isotope data of hydrogen and oxygen have been extensively utilized
in global water cycle studies (Baker et al., 2019; Bowen et al., 2019). Meanwhile, surface water
acts as a "link" between precipitation and groundwater. By integrating stable isotope data with
hydrochemical methods, researchers can gain new scientific insights into hydrological processes.
These insights include the interactions between surface water and groundwater (Yang et al., 2021;
Zhou et al., 2024), the evaporation and transpiration processes of different water bodies (Wang et
al., 2016, 2023; Xu et al., 2011), and the replenishment and infiltration of groundwater (Jasechko,
2019; Séraphin et al., 2016). Studies have shown that changes in river water isotopes can reflect
changes in local precipitation. Additionally, significant negative isotope-elevation relationships
have been observed in high mountain areas (Kong and Pang, 2016).
Reviewing past scientific studies reveals that surface water isotope data can be used as an
important tool for monitoring climate change indicators (Konecky et al., 2023; Yapiyev et al.,
2023; Zhang et al., 2023). Examples include changes in precipitation patterns and the
enhancement of evapotranspiration. These insights provide a reference basis for predicting future
climate change trends. To investigate the relationship between global surface water isotopes and
global climate, we fitted δD and δ¹⁸O data for six climate zones. The results indicated a strong
correlation between δD and δ¹⁸O across six various climate zones. The relationship between δD
and $\delta^{18}O$ for global surface water is $\delta D = 7.92\delta^{18}O + 7.80$ ($R^2 = 0.98$), which is closer to the
intercept and slope of the global meteoric water line (GMWL: $\delta D = 8\delta^{18}O + 10$), and this confirms
once again that the source of recharge of global surface water is precipitation. However, the fitted
lines of $\delta D$ and $\delta^{18}O$ for surface water were significantly different in different climatic zones (Fig.
7), and the fitted lines of $\delta D$ and $\delta^{18}O$ exhibited the lowest intercept and slope under arid climate
($\delta D = 7.50 \delta^{18}O + 3.30$, $R^2 = 0.98$), which also suggests that under arid climate, the surface water
experienced significant evapotranspiration, which led to the isotopic enrichment of surface water ,
$\delta D$ and $\delta^{18}O$ values were higher compared to other climatic zones. In the coldest polar climate
zone, the fitted line of $\delta D$ and $\delta^{18}O$ is $\delta D = 5.57\delta^{18}O + 17.18$ ($R^2 = 0.95$), and the higher slope and
intercept indicate that under the influence of the cold climate, the surface water undergoes little
evaporation, and the presence of surface water may be in the form of snow and ice, resulting in
significantly lower values of $\delta D$ and $\delta^{18}O$ compared to the other climate zones.

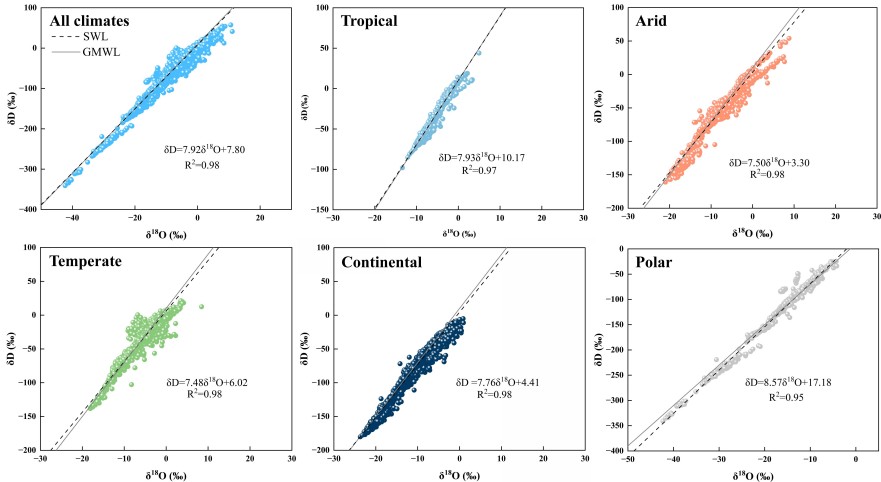

**Figure 7** Relationship between $\delta D$ and $\delta^{18}O$ in different climatic zones.
Surface water isotope data are also important for assessing ecosystem functions and
biogeochemical cycling processes (Chang et al., 2021; Chen et al., 2020). For example, by
analyzing the isotopic composition of water in rivers, lakes, or wetlands, we can understand
recharge sources, biogeochemical processes, and the ecological adaptation strategies of aquatic
organisms (Cao et al., 2022; Li et al., 2022; Zhao et al., 2024). This information provides a
scientific basis for ecosystem management and conservation. In summary, the global stable
isotope dataset of surface water offers crucial data support for our in-depth understanding of the
global water cycle, climate change, and ecosystem processes. It also aids in promoting scientific
research and sustainable development practices in related fields.

### 285    3.5 Challenges and limitations in the construction of surface water isotope
### 286    datasets and future research directions

At present, due to the limitations of sampling techniques and methods, there may be

significant differences in sampling methods and frequencies across various geological
environments and hydrogeological conditions. These differences can affect the comparison and
analysis of the data. Constructing a comprehensive isotope dataset for surface water requires
careful consideration of spatial and temporal coverage to ensure data accuracy and comparability
(Ankor et al., 2019). However, due to cost, labor, and equipment constraints, as well as the harsh
natural conditions in sampling areas, it is challenging to achieve continuous observation of
different watersheds over long time series. This limitation results in some incompleteness of the
data in terms of spatial and temporal scales (Penna et al., 2014). In addition, the accuracy of
current stable isotope data has yet to be harmonized due to issues such as sample preservation,
analytical techniques, and instrumental accuracy. These challenges may lead to problems in the
comparability and overall reliability of the data.

In the future, establishing harmonized standards for data collection, storage, and sharing will

be essential for creating a global isotope database for surface water. Additionally, integrating data
from different sources, times, and locations will be necessary to develop a more comprehensive
global isotope database for surface water (Chen et al., 2024). With advances in artificial
intelligence, there is a growing trend towards integrating isotope data with hydrologic modeling
(Gierz et al., 2017; Nelson et al., 2021). This integration promises to enhance our understanding of
hydrologic processes and improve water resource management practices. Furthermore, it
facilitates improvements in the spatial and temporal coverage of data, offering more robust
insights into water dynamics and interactions within ecosystems. Meanwhile, within the context of
global change, climate change, and isotopes are becoming increasingly integrated and
interdisciplinary. In the longer term, there is potential to develop a comprehensive understanding
and application of isotope datasets for surface water. This development will rely on integrating
expertise from disciplines such as geology, hydrology, meteorology, and others, fostering a holistic
approach to studying and managing water resources in a changing climate.



**4. Data availability**
The Global Surface Water Stable Isotope Dataset is now publicly available and the data can
be found at https://doi.org/10.17632/fs7rwp7fpr.1 (Zhu, 2024).
**5. Conclusion**
The global surface water stable isotope dataset provides crucial information for advancing
our understanding of the water cycle, climate change, and environmental monitoring. In this study,
we established a global surface water stable isotope dataset by combining measured data and
reference data from existing station data. This approach enriched the dataset and enabled
comprehensive analysis across different regions and climatic zones. The results reveal pronounced
spatial and temporal variations in the stable isotope composition of global surface water, with
significant differences observed in the isotopic composition of surface water across different
climates. The variations in global surface water isotopes are influenced by a combination of
geographic and meteorological factors, with temperature and evapotranspiration among the
climatic factors exhibiting strong explanatory power for the isotopic composition of surface water.
Observations of stable isotopes in global surface water play a crucial role in enhancing our
understanding of the global water cycle, climate change, and water resource management. They
provide essential data support for interdisciplinary research, helping to uncover connections
between hydrological processes, climate variability, and environmental changes worldwide.
Although we have enriched this dataset as much as possible, there are still regions with sparse data,
such as Siberia and Eastern Europe. In the future, efforts should focus on strengthening
observations in these challenging areas where data availability is limited. Improving the resolution
of global surface water stable isotope data can be achieved by integrating interdisciplinary
approaches and leveraging artificial intelligence methods. This approach will help fill data gaps,
enhance accuracy, and provide more comprehensive insights into global water dynamics and
environmental changes.
**ACKNOWLEDGEMENTS**
This research was financially supported by the National Natural Science Foundation of China
(42371040, 41971036), Key Natural Science Foundation of Gansu Province (23JRRA698), Key
Research and Development Program of Gansu Province (22YF7NA122), Cultivation Program of



Major key projects of Northwest Normal University (NWNU-LKZD-202302), Oasis Scientific
Research achievements Breakthrough Action Plan Project of Northwest normal University
(NWNU-LZKX-202303). We thank the editors, Nadia Ursino. The three anonymous reviewers for
providing a list of critical and very valuable comments that helped to improve the manuscript.
**Conflict of Interest Statement**
The authors declare no conflicts of interest.
**Author contributions statement**
Guofeng Zhu and Rui Li: Writing-Original draft preparation. Siyu Lu and Longhu Chen:
Data curation. Xiaoyu Qi: Writing-Reviewing and Editing. Gaojia Meng and Yuhao Wang:
Methodology. Wenmin Li: Investigation. Zhijie Zheng: Software

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
