# Peer review of "Global Stable Isotope Dataset for Surface Water 1 Rui Li1,2,3, Guofeng Zhu1,2,3\*, Longhu Chen1,2,3, Xiaoyu Qi1, Siyu Lu1,2,3, Gaojia Meng1,2,3, Yuhao 2 Wang1,2,3, Wenmin Li1,2,3, Zhijie Zheng1,2,3</"

_Earth System Science Data, 2024_

## Referee Comment (RC1)

**essd-2024-297 "Global Stable Isotope Dataset for Surface Water"**

The manuscript entitled "Global Surface Water Stable Isotope Dataset" effectively compiles stable isotope data from surface waters around the world, a first and unique dataset that is important for advancing hydrology and meteorology research. Based on my many years of work in the field of isotope hydrology, I believe this is a fundamental and important piece of work. This work will inevitably greatly promote the global sharing of data in the field of isotope hydrology and facilitate the coordinated observation of global surface water isotopes.

A sincere suggestion: the authors should consider inviting more international colleagues to participate in similar global studies. In fact, I have been following the achievements of the author's research group in recent years (which are very outstanding). I believe that globally collaborative research would be very beneficial for the authors to increase the impact of their articles, and would also better promote data sharing and global coordinated observation.

After a thorough review of the data, I am convinced that the quality control measures for the stable isotope data are rigorous and I wholeheartedly support its publication, but the authors still need to address the following questions before publication, I feel that such an excellent paper deserves better expression.

**Major comments:**

1. The introduction is well written. However, the author needs to add some research progress on stable isotope datasets for surface water and how they compare.

2. Section3.3 The predictors used here are not independent, e.g. evapotranspiration is influenced by parameters such as temperature and wind speed. Does this possible interdependence affect the results and conclusions?

3. Section 3.4 This section should highlight the applicability of the surface water stable isotope dataset, comparing it to current research and emphasizing the scientific value of the data.

4. Section 3.4 Recognize any limitations in your study that may affect the interpretation of the results.

5. Some references in the manuscript are outdated, please replace them.

**Specific comments**:

1. I apologize that I did not clearly find the location distribution of the actual points in Figure 1, the author should add this information in the text or in an additional file.

2. Why did you choose two meteorological datasets, the NCEP- NCAR reanalysis dataset and the CRUTS v. 4.07 dataset? Are they different in any way?

3. Line126: What is LIMA and is it the same as LWIA in Figure 2?

4. Line 134: The full name of ANOVA should be shown in its entirety the first time, and then the abbreviation is used in the later text, similar situation please ask the authors to solve it together.

5. Line123: Here it is $H^1/H^2$ at the beginning and δD later, are these two statements the same? Please harmonize the statements

6. Line142: What is the calculation method of RMSE and MAE?

7. Line146: 102862 should be written as 102,862, and other figures in the text should be written the same way.

8. The information in Figure 4 is vague and needs to be reformatted to improve the resolution.

9. According to Figure 1, the authors collected data for Antarctica, but this part is missing from the spatial distribution.

10. Line171: "The most pronounced variations occur in arid zones, underscoring the influence of climatic factors on stable isotopes of surface water." Are the potential causes of such pronounced variations observed in arid zones explored?

11. Line185-194: Interpolation was performed here to map changes in spatial distribution, what interpolation method was used? This should have been explained earlier.

12. Figure 5 needs to be redone to improve resolution.

13. What is meant by SWL in figure 7, which should be explained in the text?

---

## Referee Comment (RC2)

Li et al. have collected the isotopic dataset of surface water around the world via website, field observation, and reference data, which is important for understanding the hydrological processes, water resource management. Although this study analyzed these data from different aspects such as spatial and temporal scales, the relationships across different climatic zones, some analyses are too tedious and redundant. The detailed comments are listed as follows:

1.  Line 18-19, analysis, this sentence of "global surface water isotopes' spatial and temporal distribution could……" is hard to understand. Please reshape it.

2.  Line 29, research→researches.

3.  Line 41-42, "due to restrictions imposed by their conditions in various regions of the world,…" this part of sentence is very vague. Should listed in details on the conditions referring to ?

4.  Figure 2, the chart figure is too ambiguous for data processing. For example, data filtering, how you do it? From your published data, some wrong data are listed. How can explain the dashed circle data like a line. Hence, these published data should be thoroughly checked it and it is true from reference data.

[Figure]

5.  Line 49, the word of "slowly" is improper in describing the status of network.

6.  Line 243, the subtitle 3.4 is too long. I can not well catch your description on this section. It seems like a literature review and many contents are irrelevant and vague. Hence, this section should be reshaped and make it more concise.

7.  Line 81 composition of the dataset → compositions of dataset.

8.  Line 126, I can not well understand the LIMA method. How do you use this method to test raw data and ensure the data accuracy. Hence, the study should be explained it more detailed.

9.  Line 113-121, I CAN NOT well catch the author's expressions. This part is the laboratory measurements for samplers. why you state this in the manuscript.

10. Figure 2, the manuscript should be listed in detail to how do the data filtering and the reasons. For example, what motivate the data filtering for the GNIR and references data.

11. Line 133, the section of methods. This study focus on the global surface water data. Why the authors have analyzed the impact of various factors on the isotopes from Random forest model. Moreover, the motivation of this question is not well explained in the Introduction section.

12. Line 155, what means for the word "onwards".

13. Line 164. This sentence gives the seasonal variations of isotopic data. However, I DON not know this relevant information on this seasonal data including the number, location, and so on. Hence, this manuscript should include this statistical seasonal data from your website, references, and measured data. Are the current data belonged to seasonal data?

14. Line 24-256, the sentence can be removed from the manuscript. It makes the manuscript like a review manuscript. I suggest that the manuscript can be cut the similar expressions like references overlay.

---

## Author Response (AR1)

**Manuscript Number: essd-2024-297**

Revision notes: We would like to thank the editor and reviewers for the valuable and comments. which were very helpful for the further improvement of the manuscript. We have looked through each comment the editor and reviewers raised and responded and incorporate it in the revised manuscript. The following is the detailed response:

**Reviewer #1:**

The manuscript entitled "Global Surface Water Stable Isotope Dataset" effectively compiles stable isotope data from surface waters around the world, a first and unique dataset that is important for advancing hydrology and meteorology research. Based on my many years of work in the field of isotope hydrology, I believe this is a fundamental and important piece of work. This work will inevitably greatly promote the global sharing of data in the field of isotope hydrology and facilitate the coordinated observation of global surface water isotopes.

A sincere suggestion: the authors should consider inviting more international colleagues to participate in similar global studies. In fact, I have been following the achievements of the author's research group in recent years (which are very outstanding). I believe that globally collaborative research would be very beneficial for the authors to increase the impact of their articles, and would also better promote data sharing and global coordinated observation.

After a thorough review of the data,I am convinced that the quality control measures for the stable isotope data are rigorous and I wholeheartedly support its publication, but the authors still need to address the following questions before publication,I feel that such an excellent paper deserves better expression.

Response: Thank you for your support and recognition of the importance of your comments in improving the quality of our manuscript. Based on your suggestions, we have thoroughly reviewed and revised the content of the manuscript, removing non-essential repetition and making the content more concise and clear.

**Major comments:**

1. The introduction is well written. However, the author needs to add some research progress on stable isotope datasets for surface water and how they compare.

Response: Thanks to your suggestion, we have added relevant research advances to the introduction, which is revised below:

Many academics worldwide have studied the stable isotope composition of surface water, Scholars around the world for surface water stable isotope research has achieved many results, a researcher using the U.S. river water stable isotope data, mapped the isotope distribution of U.S. river water, and use the model to analyze the U.S. river water isotope changes (Bowen et al., 2011; Dutton et al., 2005).

- Bowen, G.J., Kennedy, C.D., Liu, Z., Stalker, J., 2011. Water balance model for mean annual hydrogen and oxygen isotope distributions in surface waters of the contiguous United States. J. Geophys. Res. 116, G04011. https://doi.org/10.1029/2010JG001581
- Dutton, A., Wilkinson, B.H., Welker, J.M., Bowen, G.J., Lohmann, K.C., 2005.
  Spatial distribution and seasonal variation in18 O/16 O of modern precipitation and river water across the conterminous USA. Hydrological Processes 19, 4121 4146. https://doi.org/10.1002/hyp.5876

2. Section 3.3 The predictors used here are not independent, e.g. evapotranspiration is influenced by parameters such as temperature and wind speed. Does this possible interdependence affect the results and conclusions?

Response: Random forest regression models can improve the accuracy and robustness of predictions by constructing multiple decision trees and averaging their results. In addition, random forest regression models are commonly used for prediction and classification. In this study, we used a random forest regression model to assess the importance of key meteorological variables on precipitation stable isotopes. Therefore, multicollinearity between variables is not a serious problem. A better practice for random forest regression models is to directly assess model performance and variable contributions through cross-validation and feature importance assessment (Vystavna et al., 2021).

From the smaller Root Mean Square Error (RMSE) and Mean Absolute Error (MAE), it can be seen that  $\delta^{18}$ O has a better prediction performance, while  $\delta^{2}$ H has higher RMSE and MAE values (Table S1). This is mainly due to the larger numerical range and volume of  $\delta^{2}$ H, on the other hand, these values are located in a reasonable range according to the existing studies on the prediction of stable isotopes in surface water. Therefore, we can consider the assessment of the importance of meteorological variables used in this study for the assessment of stable isotopes in surface water as reasonable and accurate.

| Variant           | RMSE  | MAE   |
|-------------------|-------|-------|
| $\delta^2 H$      | 12.87 | 10.02 |
| δ 18 Ο | 3.23  | 0.89  |

 Table S1 Random Forest Model Assessment Indicators

Vystavna, Y., Harjung, A., Monteiro, L.R., Matiatos, I., Wassenaar, L.I., 2021. Stable isotopes in global lakes integrate catchment and climatic controls on evaporation. Nat Commun 12, 7224. https://doi.org/10.1038/s41467-021-27569-x

**3. Section 3.4 This section should highlight the applicability of the surface water stable isotope dataset, comparing it to current research and emphasizing the scientific value of the data.**

Response: Thank you for your suggestions, and we have made significant adjustments to this paragraph based on the suggestions of two reviewers. We have shifted the focus to a discussion of the correlation between the surface water isotopes  $\delta^2$ H and  $\delta^{18}$ O, which helps us to better understand the role of stable isotopes in surface water as indicators of climate change. Below is the revised content:

**Global surface water $\delta^2 H$ and $\delta^{18} O$ correlations**

Here, we fit  $\delta^2$ H and  $\delta$ 18O to surface waters in six climatic zones, the results indicated a strong correlation between  $\delta^2$ H and  $\delta^{18}$ O across six various climate zones (Fig. 6). The relationship between  $\delta^2$ H and  $\delta^{18}$ O for global surface water is  $\delta^2$ H = 7.92 $\delta^{18}$ O + 7.80 (R2 = 0.98), which is closer to the intercept and slope of the global

meteoric water line (GMWL:  $\delta^2 H = 8\delta^{18}O + 10$ ), and this confirms once again that the source of recharge of global surface water is precipitation. However, the fitted lines of  $\delta^2 H$  and  $\delta^{18}O$  for surface water were significantly different in different climatic zones (Fig. 6), and the fitted lines of  $\delta^2 H$  and  $\delta^{18}O$  exhibited the lowest intercept and slope under arid climate ( $\delta^2 H = 7.50 \ \delta^{18}O + 3.30$ ,  $R^2 = 0.98$ ), which also suggests that under arid climate, the surface water experienced significant evapotranspiration, which led to the isotopic enrichment of surface water,  $\delta^2 H$  and  $\delta^{18}O$  values were higher compared to other climatic zones. In the coldest polar climate zone, the fitted line of  $\delta^2 H$  and  $\delta^{18}O$  is  $\delta^2 H = 5.57\delta^{18}O + 17.18$  (R2=0.95), and the higher slope and intercept indicate that under the influence of the cold climate, the surface water undergoes little evaporation, and the presence of surface water may be in the form of snow and ice, resulting in significantly lower values of  $\delta^2 H$  and  $\delta^{18}O$  compared to the other climate zones.

Figure 6 Relationship between  $\delta^2$ H and  $\delta^{18}$ O in different climatic zones.

**4. Section 3.4 Recognize any limitations in your study that may affect the interpretation of the results.**

Response: Thanks to your suggestion, we have revised section 3.4, which is shown in the previous section.

5. Some references in the manuscript are outdated, please replace them.

Response: Thanks to your suggestions, we have updated the older references in the manuscript.

**Specific comments**:**

1. I apologize that I did not clearly find the location distribution of the actual points in Figure 1, the author should add this information in the text or in an additional file.

Response: Thanks to your suggestion, we have added a geographic map of the actual points in the Supporting information. Below are the revised details::

Since 2015, an ecohydrological observation system has been implemented in the Shiyang River Basin in the arid zone of Northwest China to systematically gather surface water stable isotope data (Fig. S1), serving as the primary source of measured data.

Figure S1 Distribution of sampling sites in the Shiyang River Basin.

**2. Why did you choose two meteorological datasets, the NCEP-NCAR reanalysis dataset and the CRUTS v.4.07 dataset? Are they different in any way ?**

Response: CRU is one of the most widely used climate datasets currently produced by the NERC Centres for Atmospheric Science (UK) (NCAS), but the only meteorological variables included in CRUTS v.4.07 are (Precipitation rate) PRE, (Potential evapo-transpiration) PET, and (Mean 2m temperature) TEMP, while vapour pressure (VAP), wind speed (WSPD), and relative humidity (RHUM) data are obtained from the NCEP-NCAR reanalysis data were obtained.

**3. Line126: What is LIMA and is it the same as LWIA in Figure 2?**

Response: We apologize for the vague description of the Figure 2, the correct expression should be LWIA.

4. Line 134: The full name of ANOVA should be shown in its entirety the first time, and then the abbreviation is used in the later text, similar situation please ask the authors to solve it together.

Response: Thanks to your suggestion, we have filled in the full name of ANOVA here. Here are the details with modifications:

Based on previous studies, a one-way analysis of variance (ANOVA) was used to determine the significance (p